# Who are the 'social Darwinists'? On dispositional determinants of perceiving the social world as competitive jungle

Piotr Radkiewicz[1]*, Krystyna Skarżyńska[2]

1 Institute of Psychology, Polish Academy of Sciences, Warsaw, Poland, 2 Faculty of Psychology, University of Social Sciences and Humanities, Warsaw, Poland

* Piotr.Radkiewicz@psych.pan.pl

**Data Availability Statement:** All relevant data are within the manuscript and its Supporting Information files.

**Funding:** KS PR UMO-2013/09/B/HS6/03071 Polish National Science Center (NCN) UMO-2016/

## Abstract

The naive social Darwinism, also called the Competitive Jungle Belief (CJB), according to the theory of the Dual-Process Motivational (DPM) model, is recognized as an expanded perceptual scheme acting as a cognitive mediator between deep individual characteristics and the area of socio-political attitudes and ideologies. This article aims to show the individual differences that can be dispositional characteristics to believe in the Competitive Jungle scheme's principles. The presented studies' main theoretical question is to find out whether the CJB bases on positive "individual resources" or rather some psychological deficits. In an extensive survey study, including four random-representative samples of adults Poles (with N ranging from 624 to 853 respondents), we tested the predictive power of the five categories of variables: 1) attachment styles; 2) Big Five personality traits; 3) Dark Triad of personality; 4) basic human values and 5) moral judgments. The results showed the psychological profile of social Darwinists as clearly dysfunctional in terms of personal life quality. They express characteristics like admiration for power and desire to dominate, pursue one's goals at all costs, exploitative attitude towards people, and hostility. On the other hand, they reveal a fearful style in close relations with others and have low self-esteem and low self-sufficiency. From the societal perspective, such beliefs make up a vision of social life that is unfavorable for building a cooperative, helpful, and relatively egalitarian society. The supreme idea that only those who do not sympathize with others and are ready to use them can be successful and survive is far from the principles of liberal democracy.

## Introduction

Many researchers stress that individuals' and social groups' behavior are conditioned by how they perceive and understand the nature of social relations. The content of such "naive" theories about what people are and what we can expect from them activates specific actions against others. They can be prosocial (cooperation, helping) or anti-social (exploitation, manipulation, hurting) [1–7]. The social world view as a competitive jungle (also named naive social Darwinism) is probably the most complex and most straightforward expression of the negative vision

21/B/HS6/03213 Polish National Science Center (NCN) Krystyna Skarżyńska Piotr Radkiewicz.

**Competing interests:** The authors have declared that no competing interests exist.

of human nature and social relations. It directly assumes the antagonistic nature of interpersonal relations and a clear contradiction between the 'egoistic by nature' interests of individuals and social groups. This antagonism is evident on three levels that make up a consistent syndrome [8]. First, it concerns all the vital resources that we strive for in life and competes to win. They are seen as limited and impossible to multiply. There are no common benefits in such a view of the world; no good comes in the situation of cooperation. The second group of beliefs consists in attributing 'most people' aversive characteristics. People are "by nature" selfish, thoughtless, and dishonest. The last group of beliefs deals with the rules of conduct in social life—preferred, considered effective, and leading to personal success. Individuals should care solely for their good, and others must be treated ruthlessly and instrumentally. Power and money are more important than honesty and reciprocity. Cool, cynical manipulation is accepted as an effective way of achieving one's own goals. These beliefs reflect the supreme principle of naive social Darwinism—only the strongest, best adapted to life in the 'competitive social jungle' can survive.

In social psychology, the role of naive social theories sanctioning competitive social jungle principles has been highlighted by at least two influential theories. The first is the Social Dominance Theory developed by Sidanius and Pratto, which explains the emergence of hierarchical social structures based on the domination of some social groups at the expense of others [9, 10]. At the heart of the theory, the authors put social dominance orientation (SDO) recognized as a personality trait expressing a generalized desire for group domination. In this approach, Competitive Jungle Belief (CJB) can be regarded as an example of legitimizing myths (e.g., racism, sexism, class prejudices) created by dominant groups to justify social inequalities, discrimination, or anti-egalitarian policies. Thus, the Darwinian vision of the social world is not the same as SDO. They have a different status. Some aspects of Darwinian thinking serve as a mediator rationalizing/justifying high SDO in the sphere of social and political attitudes and behaviors.

The concept of CJB gained more clear psychological meaning in another influential theory called the Dual Process Motivational (DPM) model [11, 12]. Darwinian beliefs, measured with the Competitive Jungle Belief Scale, are recognized as critical determinants of high SDO in the DPM model. High SDO is activated by the perception of a social world in which social relations are captured in terms of fight, competition, and exploitation of others). Naive Darwinians view the social world as a "ruthlessly competitive jungle in which the strong win and the weak lose, which would tend to activate the motivational goals of power, dominance, and superiority over others, which in turn would be expressed in high SDO" [13]. This view is strongly related to discriminating practices towards some social groups and justification of status quo favoring social inequalities [11, 12, 14].

In the DPM model's perspective, SDO appears as a form of socio-political ideology (not personality trait), and naive Darwinian beliefs do not work only as a potential source of myths legitimizing the striving for domination. Social Darwinism becomes a form of expanded perceptual scheme, acting as a cognitive mediator that transfers environmental and personality influences into social dominance ideology, leading to many dysfunctional social phenomena. Moreover, the empirical evidence suggests its direct role in motivating people to anti-egalitarian attitudes and ideologies [12].

Thus, apart from the empirical operationalization, this approach gives the concept of social Darwinism a key theoretical status and shows its primary meaning concerning SDO. According to Duckitt and colleagues, the ideology of social dominance, contained in SDO, is one of the most important products of the naive social (colloquial) Darwinian thinking–in terms of predictive power, most powerful in predicting approval of social discrimination and inequalities.

Overall, in the context of social Darwinism, a comparison of SDO's and DPM's models theoretical potential points to the latter approach as the one that offers more inspirations for building a picture of dispositional characteristics underlying Darwinian thinking.

## The current research

As we noted in the previous section, CJB—as a cognitive mediator—links the domains of relatively stable mental dispositions and socio-political attitudes and ideologies. Based on this perspective, the potential research interests on CJB can be divided into three analytical categories: dispositional predictors, correlates (e.g., other cognitive schemas), and consequences. This paper focuses on such dispositional predictors covering personality characteristics as attachment styles, Dark Triad of personality, Big Five, basic human values, and moral judgments. We recognize them as predictors significant to understand better the core property of the social Darwinists mentality—its "anti-social" character. In other words, we assume that CJB may be the effect of not only the widespread and approved vision of the social world in a given environment but also the effect of individual propensity to strongly disturbed or directly dysfunctional relations with other people. The dispositional characteristics that we consider can play a significant role in the formation and development of individuals' acceptance of competitive jungle beliefs. They concern different personality dimensions, but all of them can push people to anti-social beliefs.

## Attachment styles

The specific conjunction of self-esteem and other people's generalized image creates a prototypical stable motivation to build interpersonal relations—the attachment style. According to Bowlby [15, 16], such prototypical attachment styles are shaped in very early childhood. They are very resistant to change in response to new adolescent and adult experiences [17]. The attachment styles play an essential role in shaping various forms of interpersonal relations and significantly affect an individual's general belief concerning human and social world nature. It means that the contents of general belief (such as the Competitive Jungle Belief) reflect the current social reality in which an individual lives and depend on stable schemata developed in the process of early socialization.

Bartholomew and Horowitz [18] argued that an individual's preferred attachment style derives from individual differences in self-image (low vs. high anxiety) and image of others (low vs. high avoidance). By combining these two dimensions, the authors obtained four theoretical attachment style prototypes: secure, dismissive, preoccupied, and fearful. Characteristics of self and others expected in CJB are contradictory with the secure attachment style, which means that an individual feels he/she deserves to be loved and be the object of interest and support. Likewise, it would be difficult to find direct link between CJB and the preoccupied style that characterize people with low avoidance (positive image of people) and high anxiety (insecure social acceptance).

Two other attachment styles—fearful and dismissive—seem to favor CJB. Both are based on the generalized negative image of people (high avoidance) that implies distrust, lack of empathy [18], suspiciousness [19], disbelief in social support and unwillingness to help [20]. Individuals with the fearful attachment style avoid social contacts and distance themselves from others, because they think that they are not worthy of care and love. They avoid close relationships for fear of being rejected or hurt, and they don't believe in social support. On the other hand, they themselves are not emphatic and not supporting. These last characteristics may result in a real deficit of social acceptance [21]. Acceptance of CJB, that is, the recognition

that the social world is bad and relationships between people are inherently antagonistic, serves to justify own failures in close relationships and anti-social reactions to rejection.

Individuals with fearful and dismissive styles are characterized by high avoidance. However, the reasons (causes) of avoidance may be different, because the fact that fearful and dismissive attachment style differ in the level of self-esteem. People with high dismissive attachment may distance themselves from others not only because they do not believe in people's good intentions but also because they feel capable of achieving their own goals without the support of others. Moreover, high but unstable self-esteem of people with dismissive style is related to narcissism. They often show hostility and aggressive behavior as a result of ego-defense reactions typical of endangered self-esteem [22]. And last but not least, because they prefer activation of the exploration system [19], and aspire to autonomy and self-reliance, they can develop some conspiracy beliefs easier than people with fearful style [23, 24].

Summarizing, the key link between CJB and attachment styles seems to be high avoidance. It is caused by social anxiety, harm or injustice (fearful attachment) and also by personal sense of self-reliance and autonomy (dismissive attachment). Thus, we expect Competitive Jungle Belief to be positively predicted by attachment styles based on high avoidance: fearful and dismissive (H1).

## Big Five personality traits

Many previous studies on the DPM model showed tough-mindedness as a personality base underlying Darwinian thinking [12]. Tough-mindedness (roughness, ruthlessness, severity) expresses the need for power, domination, and lack of empathy [25–27]. It suggests that CJB is strongly conditioned by one of the foundational Big Five personality traits—low agreeableness, described as "harsh personality" [28]. The core of low agreeableness is an antagonistic attitude towards other people and a lack of prosocial behavior. It denotes distrust, duplicity, uncompromising attitude, and unchastity [29, 30]. People who are low in agreeableness tend to be more competitive and manipulative. They take little interest in other people's problems and do not care about how others feel. They insult, belittle, and manipulate to get what they want. Thus, we suppose that the important step in building social Darwinists' psychological portrait is to determine whether the Competitive Jungle Belief is negatively related to agreeableness (H2).

## Dark Triad of personality

People who strongly believe that the social world is a competitive jungle appear anti-social, expressing aversive, antagonistic, and selfish attitudes. This general personality feature suggests direct relationships with the 'Dark Triad' of personality, which contains Machiavellianism, narcissism, and subclinical psychopathy. From a certain date, these characteristics have been studied together as three dispositional traits that are socially aversive and undesirable but not recognized as manifestations of a pathological personality [31, 32]. Machiavellianism describes an emotionally cold, egocentric, distrustful, pragmatic, cynical, and double-minded person who manipulates and controls others for personal benefits [33]. The essence of narcissism is self-absorption, high self-esteem, feeling of uniqueness and being better, arrogance, and an instrumental attitude towards people used to maintain an unrealistic self-image [34]. In turn, psychopathic individuals are characterized by a low level of anxiety and empathy, accompanied by anti-social behavior, impulsiveness, and sensation seeking [35]. In the context of naive social Darwinism, it is important to note that evolutionary psychologists consider the Dark Triad to predispose the use of highly selfish, deceptive evolutionary strategies allowing for the exploitation of partners in situations when deception is punished [36]. These behavioral

strategies are not appreciated and supported, but it may be an effective strategy from the psychobiological point of view (survival and reproductive success). Almost all studies point to systematic positive correlations among the Dark Triad components [37]. Since they are combined by their anti-social character and the deficit of empathy, we predict that Competitive Jungle Belief is positively related to Machiavellianism (H3a), narcissism (H3b), and psychopathy (H3c).

## Basic human values

The previous research suggests that people with a high level of CJB admire and desire power and dominance [12]. This characteristic should be reflected in the structure of their motivation. In the most influential current theory of personal values, Shalom Schwartz [38, 39] argues that a variety of human values express the diversity of motives by which an individual may be guided. He identified ten types of values that can be ordered on two higher-order dimensions: Openness to change *vs*. Conservation and Self-enhancement *vs*. Self-transcendence. Conservation and self-transcendence focus on social relations and interest, whereas Self-enhancement and Openness as focused on individual needs and interests. Moreover, Self-transcendence and Openness are conceptualized as anxiety-free and growth-oriented, whereas self-enhancement and conservative values are anxiety-avoidant and focused on self-protection. In terms of Schwartz's theory, social Darwinists should mostly value Self-enhancement and reject Self-transcendence. Thus we predict that Competitive Jungle Belief is positively related to Self-enhancement values (H4a) and negatively related to Self-transcendence values (H4b).

## Moral judgments

The naive Darwinian worldview cannot be neutral to moral judgments—an essential and powerful source of human motivation [40–42]. What people consider good or bad, commendable or worthy of condemnation, is reflected in their private theories of human nature and the social world. According to the descriptive and naturalistic view on morality presented in the Moral Foundations Theory [43, 44], five modular foundations are underlying human moral reasoning. Harm/care indicates basic concerns for peoples' suffering, including the virtues of caring and compassion. Fairness/reciprocity affirms fair treatment, equality, and justice. In-group/loyalty is related to group membership norms, such as loyalty, self-sacrifice, and vigilance against betrayal. Authority/respect promotes obedience and respect for authorities and is related to social order and hierarchy. The last moral fundament, purity/sanctity, concerns physical and spiritual contagion, including virtues of chastity, wholesomeness, and control of desires [43]. Two of them, Care/harm and Fairness/reciprocity, make up the Ethics of Autonomy, whereas the second dyad, In-group/loyalty, and Authority/respect, make up the Ethics of Community. Individuals, social groups, and societies put different importance on particular moral foundations. Such differences are related to ideological preferences. Research showed that lower liberalism and higher conservatism correspond to lower individualizing foundations, harm/care and fairness/reciprocity, and higher binding foundations, in-group/loyalty and authority/respect [44]. The individualizing moral foundations (the ethics of autonomy) are strong positive predictors of accepting liberal democracy's norms and values, while the Competitive Jungle Belief is not conductive to human rights' defense or care for other individuals' well-being and justice. Moreover, the naive social Darwinists are more aggressive and hostile than helpful, and they usually manipulate and compete with people than cooperate for the common good. Therefore, we predict that the Competitive Jungle Belief is negatively related to the Ethics of Autonomy expressing care/harm and fairness/reciprocity (H5).

## Materials and methods

All the research questions and hypotheses were tested in an extensive survey study project conducted within a few years on a population of adult Poles (with N ranging from 624 to 853 respondents). Due to the need to use many expanded multi-item research instruments, respondents' cognitive efficiency, and financial constraints, the project was divided into four survey studies conducted on separate samples. All surveys have been carried out by specialized research companies belonging to the European Society for Opinion and Marketing Research (ESOMAR), ensuring the observance of the highest quality and ethical standards in public opinion research and market research. The relevant committees have approved the research techniques and tools for research ethics (Institute for Social Studies, University of Warsaw, and Institute of Psychology, Polish Academy of Sciences).

### Participants and procedure

**Sample 1.** The sample consisted of 853 and was fully representative of the population in terms of sex, education, age, residence, and region. A three-stage sampling procedure was applied. In stage I, 60 territorial layers (municipality departments) based on four categories of the size of residence were drawn; in stage II, within the 60 selected layers, home addresses were drawn based on the national population registry PESEL; and in stage III, within the sampled home addresses, respondents were drawn using the Kish grid. The survey was conducted using the CAPI method.

**Sample 2.** The sample was composed of 706 respondents aged 18 to 65 years. An 'online' survey was carried out on the ARIADNA Nationwide Research Panel, with about 70,000 Polish consumers aged 15–60. Participants in the panel are subject to verification and then participate in opinion and market research receiving a small payment. Every respondent receives an individual e-mail invitation to complete the online survey. Research in this panel is conducted using the CAWI method (*Computer Assisted Web Interview*). The sample included 52.3% of females and 47.7% of males. Primary and lower education was held by 2.1% of respondents, vocational—10.3%, secondary 34.3%, post-secondary—15.2%, and 38.1% of the respondents had higher education. The overall mean age amounted to 42.1 years.

**Sample 3.** The sample consisted of 750 adults (aged 18+) questioned by the CAPI method. It was a random-quota sample. Respondents were selected based on a three-stage procedure as in Study 1 and included 52.7% of females and 47.3% of males. Primary and lower education was held by 9.5% of respondents, vocational—35.7%, secondary and post-secondary—34.2%, and 20.6% of the respondents had higher education. The overall mean age amounted to 45.4 years.

**Sample 4.** The sample was composed of 624 respondents aged 18 to 65 years. It was an online survey carried out on the ARIADNA panel using the CAWI method. It included 50.5% females and 49.5% males. Primary and lower education was held by 1.1% of respondents, vocational—8.7%, secondary 32.2%, post-secondary—26.2%, and 32.2% of the respondents had higher education. The overall mean age amounted to 37.5 years.

### Measures

**Dependent variable.** We used a 15-item Competitive Jungle Belief Scale developed by Duckitt [11] in all samples. It measures the belief that the social world is a competitive jungle characterized by a ruthless, amoral struggle for resources and power versus the belief that the social world is a place of collective harmony. Exemplary items: "My knowledge and experience tell me that the social world we live in is basically a competitive 'jungle' in which the fittest survive and succeed, in which power, wealth, and winning are everything, and might is right",

"It's a dog-eat-dog world where you have to be ruthless at times", "Life is not governed by the 'survival of the fittest', we should let compassion and moral laws be our guide (*reverse-scored)*". Internal reliability of the CJB scale ranged from .75 to .87. The exact wording of the scale is available in the S1 Appendix.

**Dispositional predictors.** *Attachment styles.* We used the Relationship Questionnaire, including a set of four vignettes constructed by Bartholomew and Horowitz [18] to measure four attachment styles: secure, dismissive, preoccupied, and fearful. They measure prototypical interaction patterns between people. Respondents were exposed to the following instruction: *'I am now going to read you several descriptions of contacts between people. Please tell me to what extent you agree that the following statements describe your typical behavior. . .'* (from 1-completely disagree to 9-completely agree).

*Big Five personality traits.* We used the NEO-Five Factor Inventory developed by Costa and McCrae [29]. It consists of 60 items (12 items on every scale) that are used to score five main dimensions of personality: Neuroticism (alpha = .87), Extraversion ($\alpha$ = .77), Openness to experience ($\alpha$ = .67), Agreeableness ($\alpha$ = .78), and Conscientiousness ($\alpha$ = .85). All items were answered using a 5-point scale format ranging from strongly disagree to agree strongly.

*Dark Triad of personality.* To measure Machiavellianism, we used the MACH-IV scale (24 items) developed by Christie and Geis [33]. Each item in the scale is a statement that must be rated as to accuracy when applied to the respondent (responses on a 7-point scale ranging from strongly agree to disagree strongly). Examples: 'It is safest to assume that all people have a vicious streak and it will come out when they are given a chance,' 'The biggest difference between most criminals and other people is that the criminals are stupid enough to get caught.' Internal reliability amounted to $\alpha$ = .71.

To measure narcissism we applied Polish adaptation [45] of the NPI questionnaire developed by Raskin and Hall [34]. It consists of 54 items with high internal reliability, $\alpha$ = .95. Responses were coded from 1 - 'This is not me' to 5 - 'This is me.' Factor analysis revealed four sub-dimensions of narcissism: Admiration demand ($\alpha$ = .89; example: 'I like to be in the spotlight'), Leadership ($\alpha$ = .91; example: 'I think I have the characteristics of a good leader'), Self-conceit ($\alpha$ = .80; example: 'I think I am special') and Self-sufficiency ($\alpha$ = .81; example: 'I always know what I am doing').

To measure psychopathy, we used a 12-item scale of psychoticism derived from the EPQ-R (S), developed by [46]. All items were answered using a dichotomous scale format (1—Yes, 2—No). Examples: 'Would you like people to be afraid of you?', 'Do good manners and tidiness matter to you?'. Internal reliability amounted to $\alpha$ = .68.

*Basic human values.* The basis of the measurement was the Portrait Values Questionnaire developed by Schwartz [38]. The applied measurement consisted of a total of 35 items including nine types of values: conformity (4 items; e.g. 'It is important to him/her always to behave properly. He/she wants to avoid doing anything people would say is improper'), tradition (5; e.g. 'Tradition is important to him/her. He/she tries to follow the customs handed down by their religion or family'), security (5; 'It is important to him/her to live in secure surroundings. He/she avoids anything that might endanger his/her safety'), self-direction (4; e.g. 'It is important to him/her to make his/her own decisions about what he/she does. He/she likes to be free and not depend on others'), universalism as caring for people (3; e.g. 'Protecting society's weak and vulnerable members is important to him/her'), benevolence (3; e.g. 'It is important to him/her to be loyal to his/her friends. He/she wants to devote himself/herself to people close to him/her'), power as dominance (3; e.g. 'He/she wants people to do what he/she says'), power as resources (3; e.g. 'He/she pursues high status and power'), and achievement (5; e.g. 'Being very successful is important to him/her. He/she hopes people will recognize his/her achievements'). Following the PVQ format, the items (descriptions) referred to an unknown person's

views and behavior. The respondent's task was to assess–on a scale from 1 (completely unlike me) to 6 (completely like me)–to what extent his/her feelings and behavior were similar to those of the presented descriptions.

In Schwartz's model, various personal values can be structuralized on two higher-order dimensions describing groups of values: Openness to change *vs*. conservation and self-enhancement *vs*. self-transcendence. In addition to the indices of particular values, two global indices measuring such higher-order preferences for values were constructed. Based on Schwartz's studies [38, 39], a list of twenty-four characteristics was completed as a result of a pilot study. They were as follows: Openness to change—autonomy, independence of thinking, curiosity about the world, inventiveness, open-mindedness, passion for discovering the world; self-enhancement—ambition, resourcefulness, effectiveness, managerial skills, successfulness, Leadership; conservation—obedience, modesty, humility, Respect for authorities, self-discipline, orderliness; and self-transcendence—helpfulness, loyalty to others, compassion for others, care for others, fairness, solidarity with others.

Every respondent received the following instruction: *In a moment, you will be presented with a set of different characteristics that everybody may possess. Generally, all of these characteristics are regarded as POSITIVE. In each set, there are six rows, including four characteristics. Please think about YOURSELF and then arrange the characteristics in each row from the one you consider the most important to the one you consider the least important TO YOU.* Next, six rows, including four characteristics, were shown to the respondent (sequentially). Each row contained one characteristic: openness to change, self-enhancement, conservation, and self-transcendence (see an example below).

The presentation of characteristics was rotated. First, twenty-four templates—invariable for the content and order of rows appearance—were prepared (see the example above). Everyone received one randomly selected template. The template included six rows with four traits. Each of the four characteristics in a single row was selected randomly from the corresponding category of values (from six possible characteristics in a first row to only one in a sixth row). A characteristic chosen first within the row was coded by a rank of 4 (as the most important), and the one chosen as the last was coded by a rank of 1 (as the least important). The applied procedure was ipsative, i.e., each rank within the row was contingent upon the other ranks in that row.

*Moral intuitions.* We used the Moral Foundations Questionnaire [44] to measure four moral intuitions: Care/harm, Fairness/reciprocity, Ingroup/loyalty, and Authority/respect. The measurement of each scale included six items: three on the subscale of moral relevance (1—not at all relevant to 6—extremely relevant) and three on the subscale of moral judgments (1—strongly disagree to 6—strongly agree). Examples: Care/Harm—'Compassion for those who are suffering is the most crucial virtue '($\alpha$ = .77), Fairness/reciprocity - 'Justice is the most important requirement for a society' ($\alpha$ = .75), Ingroup/loyalty—'It is more important to be a team player than to express oneself '($\alpha$ = .66), Authority/respect—'Respect for authority is something all children need to learn '($\alpha$ = .69).

## Results

Descriptive statistics for the measure of Competitive Jungle Belief are in the upper part of Table 1. Generally, in all samples, the distributions were shifted to the left, which means that relatively few people achieved high scores on the scale. The percentage of respondents above 1 *SD* ranged from 12% to 17%.

Detailed analysis of the relationships linking CJB with socio-demographic variables showed that naive Darwinian beliefs are somewhat more strongly revealed by males (*r* from 0.14 to

**Table 1. The scale of social Darwinism—descriptive statistics and relationships with socio-demographic variables.**

| Sample | 1 | 2 | 3 | 4 |
|---|---|---|---|---|
| | (N = 853) | (N = 706) | (N = 750) | (N = 624) |
| % > +1 SD | 17.0 | 13.0 | 15.9 | 12.0 |
| Mean | 2.53 | 2.91 | 2.93 | 2.77 |
| Standard Deviation | .56 | .67 | .68 | .72 |
| Median | 2.53 | 3.00 | 2.92 | 2.92 |
| Dominant | 2.40 | 3.53 | 3.58 | 3.50 |
| Skewness | -.07 | -.32 | -.17 | -.37 |
| Kurtosis | -.43 | .03 | -.57 | -.75 |
| | Spearman's rho coefficients | | | |
| Gender | .15** | .14** | .16** | .17** |
| Age | -.05 | -.04 | -.06 | -.19** |
| Education | -.25** | -.04 | -.02 | -.15** |
| Place of residence | -.22** | -.04 | -.09* | -.07 |

Notes.

* $p \leq 0.05$

** $p \leq 0.01$ gender coding: 1—female; 2 –male.

0.17) and by less-educated respondents ($r$ = -0.25 and -0.15 samples 1 and 4). The results also showed a decrease in Darwinian beliefs with age, although the correlation was statistically significant only in sample 4. In addition, the correlation coefficients suggest a slightly stronger tendency towards Darwinian beliefs among respondents from villages and small towns (coefficients significant in sample 1 and 3).

Table 2 presents descriptive statistics and intercorrelations for predictors from all studied samples. All the hypothetical relationships between dispositional variables and Competitive Jungle Belief were tested using multiple regression analyses. The results of subsequent testing hypotheses are presented in Table 3. We decided to put them in one table to facilitate a holistic view of the results, emphasizing measures of effect size ($\eta^2$) for predictors.

### Hypothesis 1: Attachment styles

In sample 1, we tested the predictive effects of attachment styles. The regression analysis results show that CJB was positively predicted only by the fearful attachment style (effect size $\eta^2$ = .06). It means that Darwinian beliefs tend to go hand in hand with high anxiety (negative self-image) and high avoidance (negative image of other people). These results suggest that Competitive Jungle Belief is rooted in fear of one's efficacy and life success as well as in social anxiety meaning insecurity in close, interpersonal relations. Naive social Darwinists may avoid close relationships because they do not believe that they are worthy of care and love. On the other hand, they avoid close relationships for fear of being rejected or hurt. The Competitive Jungle Belief may result from over-generalizing interpersonal traumatic experiences and may justify one's failures. In this way, naive social Darwinists can rationalize lack of life successes by ascribing human beings antagonistic personality characteristics, bad intentions, and solely selfish motivation. The experimental and survey results [47] support this interpretation: such negativistic beliefs as interpersonal distrust, belief in life as a zero-sum game, and sense of the social system's injustice characterized losers, not winners.

The H1 hypothesis was only partially confirmed because high dismissive attachment style didn't predict positively CJB. This means that the strength of the link between high avoidance

**Table 2. Descriptive statistics and zero order correlations among measured variables for all samples.**

| | | (1) | (2) | (3) | (4) | (5) | *M* | *SD* |
|---|---|---|---|---|---|---|---|---|
| *Sample 1* (*N* = 853) | | | | | | | | |
| Secure attachment style | (1) | | | | | | 5.99 | 2.24 |
| Dismissive attachment style | (2) | -.01 | | | | | 4.82 | 2.49 |
| Preoccupied attachment style | (3) | .11*** | .29*** | | | | 4.61 | 2.20 |
| Fearful attachment style | (4) | -.06 | .35*** | .42*** | | | 4.34 | 2.29 |
| *Sample 2* (*N* = 706) | | | | | | | | |
| Openness to experience | (1) | .67 | | | | | 3.17 | .41 |
| Conscientiousness | (2) | .25*** | .85 | | | | 3.59 | .52 |
| Extraversion | (3) | .22*** | .40*** | .77 | | | 3.17 | .47 |
| Agreeableness | (4) | .26*** | .44*** | .31*** | .78 | | 3.42 | .50 |
| Neuroticism | (5) | -.10** | -37*** | -.45*** | -.28*** | .87 | 2.92 | .66 |
| Machiavellianism | (1) | .70 | | | | | 4.10 | .70 |
| Psychoticism | (2) | .22*** | .68 | | | | 1.28 | .17 |
| Narcissism | (3) | .31*** | .13** | .95 | | | 2.94 | .60 |
| *Sample 3* (*N* = 750) | | | | | | | | |
| Power-resources | (1) | | | | | | 3.54 | 1.08 |
| Security | (2) | -.07* | | | | | 4.80 | .74 |
| Benevolence | (3) | -.07* | .77** | | | | 4.77 | .77 |
| Power-domination | (4) | .70*** | -.06 | -.04 | | | 3.60 | .99 |
| Openness *vs* Conservation | (5) | -.18*** | -.03 | -.05 | -.15** | | 28.3 | 6.05 |
| Self-enhance. *vs* Self-transce. | (6) | -.44*** | .22*** | .23*** | -.42*** | .48*** | 32.2 | 6.37 |
| *Sample 4* (*N* = 624) | | | | | | | | |
| Care/Harm | (1) | .84 | | | | | 4.73 | .76 |
| Justice | (2) | .85*** | .80 | | | | 4.66 | .73 |
| Loyalty | (3) | .60*** | .63*** | .73 | | | 4.18 | .69 |
| Authority | (4) | .41*** | .47*** | .75*** | .70 | | 4.01 | .70 |

*Notes.*

* $p \leq 0.05$

** $p \leq 0.01$

*** $p \leq 0.001$ Diagonals contain Cronbach's α.

attachment styles and CJB may be influenced by the differences in self-esteem. The fearful and dismissive styles share the negative perception of other people but differ radically in the self-image. We believe that due to such difference, people with a high dismissive attachment style have less motivation to take CJB as a rationalization or justification for own social failures. They may compensate for their interpersonal problems through exploration, self-reliance, autonomy and success in other fields. Their frustration is smaller than that of people with high fearful attachment. The combination of high avoidance and high anxiety leads to the search for ego defense mechanisms, and CJB may be one of them.

## Hypothesis 2: The Big Five personality traits

In this part, we tested the negative relationships between social Darwinism and agreeableness. The regression analysis, including the Big Five personality traits, resulted in the expected and remarkable negative predictive effect of agreeableness. Low agreeableness indeed very strongly favors the Competitive Jungle Belief. This relationship is incomparably greater ($\eta^2 = .27$) than all the other tested in our project. Such a strong negative relationship with agreeableness

**Table 3. Dispositional predictors of the competitive jungle beliefs—regression analyses.**

| | R | B (SE) | 95% CI | β | η² |
|---|---|---|---|---|---|
| *Sample 1* (*N* = 853) | | | | | |
| *Constant* | | 2.21 (.07)*** | (2.06; 2.35) | | |
| Secure attachment style | -.05 | -.01 (.01) | (-.03; .01) | -.04 | .00 |
| Dismissive attachment style | .16*** | .01 (.01) | (-.003; .03) | .05 | .00 |
| Preoccupied attachment style | .18*** | .02 (.01) | (-.003; .03) | .06 | .00 |
| Fearful attachment style | .34*** | .08 (.01)*** | (.05; .11) | .30 | .06 |
| $F_{(4;807)}$ = 21.8*** $R^2$ = 0.10 | | | | | |
| *Sample 2* (*N* = 706) | | | | | |
| *Model 1* | | | | | |
| *Constant* | | 6.33 (.28)*** | | | |
| Openness to experience | -.30*** | -.26 (.05)*** | (-.36; -.16) | -.16 | .04 |
| Conscientiousness | -.36*** | -.19 (.05)*** | (-.28; -.10) | -.15 | .02 |
| Extraversion | -.11** | .19 (.05)*** | (.09; .29) | .13 | .02 |
| Agreeableness | -.59*** | -.72 (.05)*** | (-.81; -.63) | -.54 | .27 |
| Neuroticism | .13*** | -.03 (.03) | (-.09; .04) | -.03 | .00 |
| $F_{(5;705)}$ = 93.2*** $R^2$ = 0.40 | | | | | |
| *Model 2* | | | | | |
| *Constant* | | .70 (.20)*** | | | |
| Machiavellianism | .44*** | .32 (.03)*** | (.26; .38) | .34 | .14 |
| Psychopathy | .42*** | 1.08 (.12)*** | (.84; 1.33) | .27 | .10 |
| Narcissism: Admiration demand | .29*** | .19 (.05)*** | (.10; .29) | .21 | .02 |
| Narcissism: Leadership | .18** | .16 (.05)** | (.06; .26) | .17 | .01 |
| Narcissism: Self-conceit | .10* | -.07 (.04) | (-.14; .00) | -.08 | .00 |
| Narcissism: Self-sufficiency | -.15** | -.38 (.04)*** | (-.46; -.29) | -.35 | .09 |
| $F_{(6;705)}$ = 74.8*** $R^2$ = 0.39 | | | | | |
| *Sample 3* (*N* = 750) | | | | | |
| *Model 1* | | | | | |
| *Constant* | | 3.86 (.15)*** | (3.57; 4.16) | | |
| Power-resources | .46*** | .20 (.02)*** | (.15; .25) | .32 | .08 |
| Security | -.45*** | -.22 (.04)*** | (-.30; -.14) | -.25 | .04 |
| Benevolence | .44*** | -.20 (.04)*** | (-.27; -.12) | -.23 | .04 |
| Power-domination | .40*** | .11 (.03)*** | (.05; .16) | .16 | .02 |
| $F_{(4;745)}$ = 133.4*** $R^2$ = 0.42 | | | | | |
| *Model 2* | | | | | |
| *Constant* | | 4.12 (.13)*** | (3.87; 4.38) | | |
| Openness *vs* Conservation | -.10** | .01 (.004)** | (.005; .022) | .12 | .01 |
| Self-enhance. *vs* Self-transc. | -.40*** | -.05 (.004)*** | (-.06; -.04) | -.46 | .16 |
| $F_{(2;747)}$ = 76.3*** $R^2$ = 0.17 | | | | | |
| *Sample 4* (*N* = 624) | | | | | |
| *Constant* | 4.72 (.17)*** | (4.38; 5.05) | | | |
| Care/Harm | -.55*** | -.53 (.06)*** | (-.65;-.42) | -.57 | .12 |
| Justice/Reciprocity | -.47*** | -.15 (.06)** | (-.27; -.03) | -.15 | .01 |
| In-group/Loyalty | -.18** | .11 (.06) * | (-.01; .22) | .09 | .00 |
| Authority/respect | -.02 | .21 (.05)*** | (.11; .31) | .20 | .03 |

(*Continued*)

**Table 3.** (Continued)

|  | *R* | *B* (*SE*) | 95% CI | β | η² |
|---|---|---|---|---|---|
| $F_{(4;619)} = 87.1^{***}$ $R^2 = 0.36$ |  |  |  |  |  |

*Notes.*

** $p \leq 0.01$

*** $p \leq 0.001$.

*r*—zero order correlation r-Pearson coefficients; *B*—unstandardized regression coefficients

*SE*—standard error of *B*; 95% CI—confidence intervals for *B*; β—standardized regression coefficients

η²—partial eta squared as an effect size measure.

confirmed earlier results reported by Duckitt and colleagues [11]. We also observed a small positive predictive effect of extraversion ($\eta^2 = .02$) and small but negative effects of openness to experience and conscientiousness ($\eta^2 = .04$ and .02, respectively).

## Hypothesis 3: The Dark Triad of personality

The analysis concerning the Dark Triad of personality resulted in expected substantial and positive relationships with Machiavellianism and psychopathy ($\eta^2 = .14$ and .10, respectively). These results again demonstrate antisocial inclinations and the deficit of empathy as typical for social Darwinists. As to the hypothesis concerning narcissism, the four-element measurement of narcissism revealed positive (though very moderate) predictive effects of the need for admiration and leadership ($\eta^2 = .02$ and .01, respectively). However, self-conceit was non-significant, and the effect of self-sufficiency turned out to be significant but negative ($\eta^2 = .09$). This finding suggests that Darwinian thinking is not based on a strong faith in one's abilities and in succeeding through oneself. Instead, social Darwinists rather strive to pursue their goals by exploiting the resources of other people.

As a personality trait, Machiavellianism describes manipulative individuals who deceive and exploit others to achieve personal goals. People high in Machiavellianism have a dark picture of human nature and are inclined to adopt aggressive social strategies in response to differing social and economic conditions [33, 48]. Machiavellian personality seems to be one of the most substantial dispositional bases of the Competitive Jungle Belief. Psychopathy, as another type of antisocial personality characteristic, also predicts CJB. It is characterized by a lack of empathy and compassion and facilitates approval for harming and exploiting individuals or social groups [49, 50]. "The dark personality" leads people to the acceptance of some behavioral aspects of CJB—caring only for the own material good or psychological well-being and treating others ruthlessly and instrumentally. Cool, cynical manipulation is accepted by people with psychopathic personalities and treated as an effective way of achieving their own goals.

The role of the third type of "Dark Triad of personality"—narcissism—is not so univocal. On the one side, people high in narcissism demand admiration and leadership; these aspects of narcissism are significant positive predictors of CJB. On the other side, high narcissistic self-sufficiency is CJB's considerable negative predictor. This finding suggests that Darwinian thinking is not based on a strong faith in one's abilities and in succeeding through oneself. Instead, social Darwinists rather strive to pursue their goals by exploiting the resources of other people.

## Hypothesis 4: Basic human values

In this part of our study, we tested the axiological predictors of the CJB level in the pool of basic personal values proposed by Schwartz. Table 3 presents two separate regression analyses:

1) including standard indices of the measured values, and 2) including two higher-order dimensions: self-enhancement *vs.* self-transcendence and openness to change *vs.* conservation. Due to the number of predictors and some robust correlations between them, in the first regression analysis, we used the stepwise selection of variables (in order to leave only statistically significant predictors). In line with our expectations, the results showed that—on the higher-order level—the CJB was strongly related to the preference for self-enhancement values and rejection of self-transcendence values ($\eta^2 = .12$). The results also point to marginal positive relationship with conservation values and marginal positive relationship with openness to change values ($\eta^2 = .01$). On the lower-order level, the CJB scale revealed strong positive link with striving for resources and smaller link with domination ($\eta^2 = .08$ and $.02$, respectively). Apart from the relationships with both aspects of power, we also observed noticeable negative links with benevolence and security ($\eta^2 = .04$ and $.04$, respectively).

### Hypothesis 5: Moral judgments

As the results in Table 3 show, in line with our expectations, Darwinian beliefs turned out to be in opposition to the ethics of Autonomy. We observed a strong negative relationship with care/farm ($\eta^2 = .12$), accompanied by a much smaller but significant negative relationship with justice/reciprocity ($\eta^2 = .01$). We also observed a noticeable positive predictive effect of authority/respect that expresses a preference for hierarchical social relations based on power and subordination to authority ($\eta^2 = .03$). These results are concordant with value preferences characteristic for individuals with high Competitive Jungle Beliefs (stress on power-resources and power-domination).

## Summary and discussion

We aimed to show people's psychological profile believing in such principles of the competitive social jungle as "struggle for existence" or "survival of the fittest." By the term, Competitive Jungle Belief (or Social Darwinism) we mean a specific naive theory of the social world. It belongs to the broader category of social beliefs, whose common denominator is a profoundly pessimistic and negativistic view of human nature and interpersonal relations. We believe two singularities constitute the uniqueness of this construct. The first one lies in a specific, pseudo-scientific foundation based on the straightforward application of evolutionary principles to the human population. The second one lies in the CJB's "directivity," as this way of thinking gives an individual a clear set of conduct and personal success rules.

In our research, we used the theoretical perspective of the Dual-Process Motivational model, which argues that the Competitive Jungle Belief works as a cognitive structure mediating between deep individual characteristics and socio-political area attitudes and ideologies. This perspective opens a broad area for research on CJB. Earlier studies focused mainly on consequences of CJB, demonstrating some reasons why social Darwinism can be troublesome for a society [11–13, 51, 52]. The CJB turned out to favor anti-egalitarian ideologies and attitudes—measured as social dominance orientation or economic conservatism. Both mentioned variables imply hierarchical social relations and legitimize economic and social inequalities. CJB also mediates between dispositional aggressiveness and approval of aggression in social and political life.

As CJB may lead to some dysfunctional social phenomena, we believe that recognizing its dispositional sources needs special attention. Therefore, we realized the project in which psychological predisposition to acceptance or rejection CJB was searched for. Our results show Darwinian thinking as a phenomenon related to numerous personality antecedents. Its most expressive characteristics are low agreeableness and the components of the Dark Triad of

personality. These characteristics describe people with a generalized antagonistic attitude towards others. Low agreeableness means a lack of prosocial behavior, distrust, duplicity, inability to compromise. The Dark Triad of personality is a multi-faceted constellation of "anti-communal" characteristics (selfishness, distrust, hypocrisy) and missing communal characteristics (empathy, altruism, compassion).

On the one hand, such characteristics are based on pure antagonism and lack of respect for social norms. On the other hand, they express the tendency to exploit and seek admiration [32]. Among the Dark Triad components, the strongest predictor of CJB turned out to be Machiavellianism, characterized by emotional coldness, cynicism, and insensitivity to others' feelings. This generalized antisocial attitude is also reflected in noticeable psychopathic inclinations revealed in the form of hostile feelings (suspiciousness, jealousy, obstinacy).

We expected that the main noticeable features of social Darwinists' personality should be reflected in the relevant set of personal values. Furthermore, the results showed a full coherence between the sphere of Darwinian personality and underlying axiological fundaments. No doubt, perceiving the social world as a competitive jungle is primarily related to self-expansion values that come from exercising power, domination, accumulation of resources, and achieving personal success. Such a personality picture and hierarchy of values need compatible and supportive moral judgments. Indeed, we found that the configuration of moral intuitions owned by social Darwinists is very in line with their personality and personal values. In terms of the Moral Foundations Theory [44], they are 'sinners' against the care/harm and justice/reciprocity moral codes. They especially reject human rights, care, help, and compassion as relevant criteria of moral judgments. As a matter of fact, they did not value any measured ethical code. If they do, it is some respect for authorities. The only noticeable premise for considering something good or bad is the power of authority and social relations hierarchy.

So far, the psychological picture of naive social Darwinist seems relatively coherent and unambiguous. Its basic features are admiration for power, desire to dominate, the pursuit of one's goals at all costs, antisocial, and exploitative attitude towards people. One would also expect high self-esteem, self-confidence, and self-sufficiency in relations with people. Meanwhile, it turns out that growing social Darwinism is related to low, not high, self-esteem. Our results showed a positive relationship between CJB and the fearful attachment style. According to Bartholomew and Horowitz [18], the individual's preferred attachment style derives from individual differences in self-image and other-image. Social Darwinists' tendency to express the fearful attachment style has two interpretative consequences. First, and once again, it suggests the negative image of other people—this is reflected in high avoidance, which means unwillingness to get close to others and be dependent on someone. On the other hand, the fearful attachment style is also determined by high anxiety, meaning low self-acceptance and fear of rejection. As noticed by Leary and colleagues [21], social Darwinists' self-esteem is a reaction to social disapproval and lack of recognition [53]. This conclusion is probably in line with the observed elevated level of hostility if we assume that hostility of narcissist persons usually draws from the sense of being depreciated [54]. In sum, these results suggest that people with high CJB avoid intimate relationships for fear of being rejected or hurt, and they tend to think they are not worthy enough of care and love. Their striving for power and domination does not seem to result from a sense of power and self-respect. More speaks for the fact that it is a form of compensatory/defensive strategy.

The Competitive Jungle beliefs about the social world are a phenomenon that we can analyze both on the societal and individual level. When it concerns the social one, it is rather unquestionable that Darwinian beliefs make up a vision of social life that is unfavorable for building a cooperative, helpful, and relatively egalitarian society. Cool, cynical manipulation and even some forms of violence are accepted as effective ways of achieving one's own goals. It

is not a pro-democratic inclination. Liberal democracy, as the idea and as the political system, postulates maximizing well-being for all members of society, minimizing violence, and promoting human rights [55, 56]. Democratic policies aim to eliminate the "principle of violence" in internal and international relations [56, 57]. The supreme idea that *only those who do not sympathize with others and are ready to 'use' them can be successful and survive in the social jungle* is undoubtedly far from democratic principles.

The competitive jungle belief is conflicting with the above understanding of democracy. However, it can support adversarial democracy [58, 59], the type of politics that aims to gain an advantage over the opponents and destroy them politically—to deprive them of power, good reputation, and economic strength. The adversarial democracy's psychological essence is the belief that a ruthless struggle for power among political parties and individual politicians is good for the public interest. However, it mainly undermines the efficiency of democratic politics and destroys democratic communities [60].

At the individual level, the most expressive characteristic of naive social Darwinism is the multi-faceted antagonistic attitude towards other people. Besides, we noticed something like a mental split underlying this thinking—the worship and admiration for strength and power coexist with a somewhat fragile and uncertain self-image. The fearful attachment style observable in naive social Darwinists may promote a competitive social jungle vision supposed to have a compensatory/defensive function. However, the compensatory/defensive strategy in the form of a vicious circle may induce a more profound sense of alienation or social rejection.

## Limitations and future research

We have devoted this project to the dispositional determinants of social Darwinism. The choice of this and no other form of beliefs about the social world was dictated by the fact that the social and political consequences of CJB seemed to us to be particularly detrimental to social cohesion and the democratic order. Nevertheless, Darwinian beliefs, even if they are particularly harmful, are only one manifestation of a broader category of phenomena that may be called negativistic schemes of the social world.

Such schemes are important category in the colloquial account of social reality. They imply the belief that the nature of interpersonal relations is antagonistic and the interests of various individuals and social groups ("egoistic by nature") are incompatible democracy [4, 7, 8, 59–61]. Apart from CJB, the negativistic schemes of the social world include, among others: belief in dangerous and threatening world [62], belief in life as a zero-sum game [47], and generalized interpersonal distrust [63]. They all share a pessimistic view of the human nature and interpersonal relations.

All these sets of beliefs are negativistic, albeit with partly different correlates, and activate different motivational goals. For example, seeing the social world as dangerous and threatening activates the motive of social control and security (which should lead to increased authoritarianism), while a competitive-jungle worldview activates the power motive and should lead to an enhanced social dominance orientation (SDO). So far, we still know too little about the negative social beliefs' psychogenesis and mutual relationships between them. Factors conditioning such beliefs have been sought mainly from experiences of individuals, groups, and entire communities [61, 64–66] as well as from some personality traits and stable evaluative orientations [22–24, 67].

Presumably, negativistic schemes of the social world undermine the foundations of collective life—the ability to cooperate and compromise, interpersonal trust, the ability to formulate and achieve collective goals and many others. Therefore, in further research, we should pay more attention to social processes that replicate and multiply the negative view of the social

world. For example, it is possible that political conflicts and various forms of social polarization result in the cognitive integration of all these negativistic schemas into a single generalized worldview. The salient changes in individual and social life lead to accepting certain beliefs and refusing the others. As our research shows, in the pre-pandemic times, the average level of CJB in the society was relatively low. During the Covid-19 pandemic, a considerable increase in social negativism can be expected (like Darwinian thinking and the belief in social world's threatening nature). This may be a very good starting point for further research aimed at formulating a broader theory of negativistic thinking about the social world. As in the case of the belief in competitive jungle and the belief in dangerous world in the Dual-Process model, it could indicate similar attitudinal and behavioral consequences determined by different motivational needs and personality dispositions.

## Conclusions

The Competitive Jungle beliefs about the social world are a phenomenon that we can analyze both on the societal and individual level. When it concerns the social one, it is rather unquestionable that Darwinian beliefs make up a vision of social life that is unfavorable for building a cooperative, helpful, and relatively egalitarian society. Cool, cynical manipulation and even some forms of violence are accepted as effective ways of achieving one's own goals. It is not a pro-democratic inclination. Liberal democracy, as the idea and as the political system, postulates maximizing well-being for all members of society, minimizing violence, and promoting human rights [55, 56]. Democratic policies aim to eliminate the "principle of violence" in internal and international relations [56, 57]. The supreme idea that *only those who do not sympathize with others and are ready to 'use' them can be successful and survive in the social jungle* is undoubtedly far from democratic principles.

The competitive jungle belief is conflicting with the above understanding of democracy. However, it can support adversarial democracy [58, 59], the type of politics that aims to gain an advantage over the opponents and destroy them politically—to deprive them of power, good reputation, and economic strength. The adversarial democracy's psychological essence is the belief that a ruthless struggle for power among political parties and individual politicians is good for the public interest. However, it mainly undermines the efficiency of democratic politics and destroys democratic communities [60].

At the individual level, the most expressive characteristic of naive social Darwinism is the multi-faceted antagonistic attitude towards other people. Besides, we noticed something like a mental split underlying this thinking—the worship and admiration for strength and power coexist with a somewhat fragile and uncertain self-image. The fearful attachment style observable in naive social Darwinists may promote a competitive social jungle vision supposed to have a compensatory/defensive function. However, the compensatory/defensive strategy in the form of a vicious circle may induce a more profound sense of alienation or social rejection.

## Supporting information

**S1 File. Study 1 dataset.**
(XLS)

**S2 File. Study 2 dataset.**
(XLSX)

**S3 File. Study 3 dataset.**
(XLSX)

**S4 File. Study 4 dataset.**
(XLSX)

**S1 Appendix.**
(DOCX)

## Author Contributions

**Conceptualization:** Piotr Radkiewicz, Krystyna Skarżyńska.

**Data curation:** Piotr Radkiewicz.

**Formal analysis:** Piotr Radkiewicz.

**Funding acquisition:** Piotr Radkiewicz, Krystyna Skarżyńska.

**Investigation:** Piotr Radkiewicz, Krystyna Skarżyńska.

**Methodology:** Piotr Radkiewicz.

**Project administration:** Krystyna Skarżyńska.

**Supervision:** Piotr Radkiewicz, Krystyna Skarżyńska.

**Validation:** Krystyna Skarżyńska.

**Writing – original draft:** Piotr Radkiewicz, Krystyna Skarżyńska.

**Writing – review & editing:** Piotr Radkiewicz, Krystyna Skarżyńska.

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
