## [Decision Letter · Decision Letter 0]

30 Apr 2021

PONE-D-21-02231

Who are the „Social Darwinists’? On dispositional determinants of perceiving the social world as competitive jungle

PLOS ONE

Dear Dr. Radkiewicz,

Thank you for submitting your manuscript to PLOS ONE. After careful consideration, we feel that it has merit but does not fully meet PLOS ONE’s publication criteria as it currently stands. Therefore, we invite you to submit a revised version of the manuscript that addresses the points raised during the review process. Reviewers pointed some additional statistical (e.g., reporting zero-order correlations, dealing with shared variance) and editorial suggestions (e.g., being more explicit about limitations) that I think will be rather easy to address and hope you can so soon. Please submit your revised manuscript by Jun 14 2021 11:59PM. If you will need more time than this to complete your revisions, please reply to this message or contact the journal office at plosone@plos.org. Please include the following items when submitting your revised manuscript:

We look forward to receiving your revised manuscript.

Kind regards,

Peter Karl Jonason

Academic Editor

PLOS ONE

Journal Requirements:

2. You indicated that ethical approval was not necessary for your study. We understand that the framework for ethical oversight requirements for studies of this type may differ depending on the setting and we would appreciate some further clarification regarding your research. Could you please provide further details on why your study is exempt from the need for approval and confirmation from your institutional review board or research ethics committee (e.g., in the form of a letter or email correspondence) that ethics review was not necessary for this study? Please include a copy of the correspondence as an ""Other"" file.

3. Please note that PLOS ONE does not copy edit accepted manuscripts (https://journals.plos.org/plosone/s/criteria-for-publication#loc-5). To that effect, please ensure that your submission is free of typos and grammatical errors.

Reviewers' comments:

Reviewer's Responses to Questions

**Comments to the Author**

1. Is the manuscript technically sound, and do the data support the conclusions?

Reviewer #1: Yes

Reviewer #2: Yes

2. Has the statistical analysis been performed appropriately and rigorously? 

Reviewer #1: Yes

Reviewer #2: Yes

3. Have the authors made all data underlying the findings in their manuscript fully available?

Reviewer #1: Yes

Reviewer #2: Yes

4. Is the manuscript presented in an intelligible fashion and written in standard English?

Reviewer #1: Yes

Reviewer #2: Yes

5. Review Comments to the Author

Reviewer #1: This is an interesting manuscript that investigates the major individual difference predictors of a Competitive Jungle Worldview (CJW). The samples are impressive and overall it makes I think a significant contribution to the research literature and merits publication. I do, however, have several suggestions for revision that should be easily accomplished and should enhance the manuscript:

1. A table giving the zero order correlations between all the predictor variables and CJW (including socio-demographic) would be useful and also helpful in interpreting the regression effects.

2. The samples are very large so that virtually all effects become statistically significant. I’d suggest that in such cases conventional significance is less useful and it would be better to either tighten that to .01 or even better to use some effect size convention, such as Cohen’s to decide on what effects are practically significant and meriting discussion.

3. It seems very likely that the four Narcissism facets would be extremely highly intercorreated and thus highly likely to distort the Betas rendering them very likely to give misleading effects. If they are indeed very highly correlated (i.e., >.50) I’d suggest using only the single Narcissism score in the regression in Table 2, Sample 2, Model 2.

4. Table 2 needs notes giving the meaning of all statistical symbols used in the table.

Reviewer #2: This is a very interesting manuscript, well written and conceptually important. The main problem I see in the design, is that dangerous world beliefs (DWB, the other worldview posited by Duckitt’s model) was apparently not measured. This is regrettable because a richer picture would have emerged reasoning on the discriminant associations of the measured antecedents with CJB vis a vis DWB. At the very least, the authors should point this a serious limitation, and discuss theoretically what could have been expected from the associations of the antecedents considered and DWB.

A second, less important concern, is that clear to me why the authors outrightly dismiss as an antecedent of CJB the dismissive (high avoidance, low anxiety) attachment style. And why attach the fearful style to CJB instead. From my perspective, fear and related cognitive and emotional processes seems to fit better with dangerous world belief rather than with CJB. I think the developmental reasoning behind Dickitt’s DPM points in this direction too. The Avoidant component of dismissive attachment entails lack of empathy and suspiciousness. It also entails a suppressive style of negative emotional experiences and a linked tendency to projecting negative emotionality and negative intentions onto others. Empirically, this could translate in adopting dysfunctional and socially costing beliefs as conspiracy beliefs (Leone et al., 2019). It appears to me that such features fit well with adopting a CJ worldview. The rationale expounded by the authors (e.g., preventing negative social experience, low self-regard, etc.) can be also linked with Avoidant and dismissive styles. If the authors would elaborate more on their positions, I think they should clarify if and how CJB and DWB differ in terms of attachment styles. A dismissive style is also associated with lower levels of self-reported anxiety, which is in turn consistent with the rationale you offer for linking the dark triad to CJB (page 7).

Results. It would be important to report zero order correlations among measured variables for all samples, not confining such information to supplementary materials. I cannot understand what asterisks stand for in Table 2 (sample 1), because they appear close to bs both significant and non significant (judging from the 95% CIs).

A final suggestion is to consider the possibility of moving to a latent variable level: this would increase power and provide more precise estimates of the effect sizes. This is important because differences in reliabilities of the measured antecedents are not negligible.

As I mentioned above, limitations should be explicitly considered and discussed.

Leone, L., Giacomantonio, M., Williams, R., & Michetti, D. (2018). Avoidant attachment style and conspiracy ideation. Personality and Individual Differences, 134, 329-336.

6. PLOS authors have the option to publish the peer review history of their article (what does this mean?). If published, this will include your full peer review and any attached files.

Reviewer #1: No

Reviewer #2: No

---

## [Author Response · Author response to Decision Letter 0]

22 Jun 2021

Dear Editor and Reviewers,

Thank you for good news about the submitted paper. Also many thanks to Reviewers for their remarks and comments. Along with the revised text we’re attaching the following description of what has been done. 

with best regards, 

Piotr Radkiewicz and Krystyna Skarżyńska

Reviewer #1:

1. A table giving the zero order correlations between all the predictor variables and CJW (including socio-demographic) would be useful and also helpful in interpreting the regression effects.

The column giving the zero order correlations between all the predictor variables has been attached to Table 2. Besides, we have included the CJB’s correlations with socio-demographic variables in the lower part of Table 1

2. The samples are very large so that virtually all effects become statistically significant. I’d suggest that in such cases conventional significance is less useful and it would be better to either tighten that to .01 or even better to use some effect size convention, such as Cohen’s to decide on what effects are practically significant and meriting discussion.

We did it. In the results section, we refer not to the regression coefficients, but to the eta squared coefficients, which is the recommended measure of the effect size in regression analyses

3. It seems very likely that the four Narcissism facets would be extremely highly intercorrelated and thus highly likely to distort the Betas rendering them very likely to give misleading effects. If they are indeed very highly correlated (i.e., >.50) I’d suggest using only the single Narcissism score in the regression in Table 2, Sample 2, Model 2.

This is a very accurate point. The facets of narcissism are indeed strongly correlated - ranging from .45 to .70. However, replacing them with the global narcissism measure does not seem to be a good solution to us, because it significantly lowers the fit of the model (the global narcissism effect is Beta = .02 and is non-significant). So we can distort the effect of narcissism in this way and lose a lot of interesting information. We performed separate regression analyzes with single components of narcissism, which showed the following beta effects: Self-sufficiency = -0.20, Admiration-demand = 0.14, Leadership = 0.05, and Self-conceit = -0.01. This shows that the coefficients in the model with four narcissism components (Table 3) are not misleading but rather extract the actual magnitudes of individual effects (which are supressed in the global index due to reverse relationships).

4. Table 2 needs notes giving the meaning of all statistical symbols used in the table.

Notes for all symbols were added (now it is Table 3) 

Reviewer #2:

The main problem I see in the design, is that dangerous world beliefs (DWB, the other worldview posited by Duckitt’s model) was apparently not measured. This is regrettable because a richer picture would have emerged reasoning on the discriminant associations of the measured antecedents with CJB vis a vis DWB. At the very least, the authors should point this a serious limitation, and discuss theoretically what could have been expected from the associations of the antecedents considered and DWB.

According to DPM model, the personality dimension underlying CJB is tough-mindedness. It implies striving for power and pursuing one's own interests at all costs, and results in SDO. On the other hand, the Dangerous World Belief (DWB) expresses the value of establishing and maintaining collective security, order, stability, and group cohesion - as opposed to freedom, autonomy or self-expression. DWB’s goals and beliefs follow from personal disposition to social conformity (as opposed to autonomy). Not only studies on the DPM model, but also some other studies have shown different sets of CJB's and DWB's predictors. However, DWB was not studied in our research so far because it was not our goal to provide another extension of the DPM model. Our goal was to identify available antecedents of CJB, because it is this type of social beliefs that we find most dangerous for the democratic order and social cohesion. Although we have consciously not contrasted CJB with DWB so far, the Reviewer is right that showing the similarities and differences in the psychological foundations of CJB and DWB would be an important supplement to our research. It would also be important to show the circumstances in which DWB and CJB stimulate each other, possibly accumulating their negative effects. We’re writing about it in the new sub-section titled Limitations and future research, which was introduced as suggested by the Reviewer.

A second, less important concern, is that clear to me why the authors outrightly dismiss as an antecedent of CJB the dismissive (high avoidance, low anxiety) attachment style. And why attach the fearful style to CJB instead. From my perspective, fear and related cognitive and emotional processes seems to fit better with dangerous world belief rather than with CJB. I think the developmental reasoning behind Dickitt’s DPM points in this direction too. The Avoidant component of dismissive attachment entails lack of empathy and suspiciousness. It also entails a suppressive style of negative emotional experiences and a linked tendency to projecting negative emotionality and negative intentions onto others. Empirically, this could translate in adopting dysfunctional and socially costing beliefs as conspiracy beliefs (Leone et al., 2019). It appears to me that such features fit well with adopting a CJ worldview. The rationale expounded by the authors (e.g., preventing negative social experience, low self-regard, etc.) can be also linked with Avoidant and dismissive styles. If the authors would elaborate more on their positions, I think they should clarify if and how CJB and DWB differ in terms of attachment styles. A dismissive style is also associated with lower levels of self-reported anxiety, which is in turn consistent with the rationale you offer for linking the dark triad to CJB (page 7).

Reviewer is absolutely right to note that both groups, people with fearful attachment style and people with dismissive attachment style, are characterized by a generalized negative image of other people, resulting from the frustrated need for closeness. So, we agree that defensive avoidant component is present not only in the fearful style, but also in the dismissive style. It is also true that both styles “can entail a suppressive style of negative emotional experiences and a linked tendency to projecting negative emotionally or negative infusions into others.” Therefore, we have modified the H1 hypothesis in line with the Reviewer’s comments. However, ultimately, controlling for the mutual covariance, only the fearful style, not the traumatic one, turned out to be a clear positive predictor of Darwinian thinking. We present our interpretation of this empirical fact in the appropriate part of the Results section. 

Results. It would be important to report zero order correlations among measured variables for all samples, not confining such information to supplementary materials.

The table with zero order correlations was added. Now it is Table 2 

I cannot understand what asterisks stand for in Table 2 (sample 1), because they appear close to bs both significant and nonsignificant (judging from the 95% CIs).

This was corrected, thank you 

A final suggestion is to consider the possibility of moving to a latent variable level: this would increase power and provide more precise estimates of the effect sizes. This is important because differences in reliabilities of the measured antecedents are not negligible.

We chose not to replace the regression models with latent variable models. Although it was only the Reviewer’s suggestion, we did some preliminary attempts to use the latent variable model. However, it turned out that due to the ratio of the sample size to degrees of freedom, some of the tested models would not meet the minimum power requirements. For example, in model 1 for sample 2, we have 75 items. Where n = number of items, we have n(n+1)/2 sample moments. In this case it is 2850 bits of information. On the other hand, we have 75 observed variable variances, 69 factor loadings (because one item from each latent construct to one of its manifest indicators is needed to "set the metric" of the latent construct), 6 latent variable variances, 5 regression paths in the model, and 21 latent variable covariances between the exogenous constructs. This gives 176 parameters to be estimated and 2674 degrees of freedom. And this gives 4.01 (706/176) respondents per estimated parameter which is definitely below 5 recommended as the low end of acceptability for power. 

As I mentioned above, limitations should be explicitly considered and discussed.

We have added a separate section on limitations and future research, thank you again.

---

## [Editor Report · Decision Letter 1]

28 Jun 2021

Who are the „Social Darwinists’? On dispositional determinants of perceiving the social world as competitive jungle

PONE-D-21-02231R1

Dear Dr. Radkiewicz,

We’re pleased to inform you that your manuscript has been judged scientifically suitable for publication and will be formally accepted for publication once it meets all outstanding technical requirements.

Kind regards,

Peter Karl Jonason

Academic Editor

PLOS ONE
---

## [Editor Report · Acceptance letter]

16 Jul 2021

PONE-D-21-02231R1 

Who are the „Social Darwinists’? On dispositional determinants of perceiving the social world as competitive jungle 

Dear Dr. Radkiewicz:

I'm pleased to inform you that your manuscript has been deemed suitable for publication in PLOS ONE. Congratulations! Your manuscript is now with our production department. 

Kind regards, 

on behalf of

Dr. Peter Karl Jonason 

Academic Editor

PLOS ONE